# Costs of Employee Stewardship Behaviors for Employees in the Work-to-Family Penetration Context during the COVID-19 Pandemic

**DOI:** 10.3390/ijerph19106117

**Published:** 2022-05-18

**Authors:** Chen Qian, Xinran Gu, Lei Wang

**Affiliations:** 1School of Business Administration, South China University of Technology, Guangzhou 510640, China; 201710105492@mail.scut.edu.cn (C.Q.); bmguxinran@mail.scut.edu.cn (X.G.); 2School of Politics and Public Administration, South China Normal University, Guangzhou 510006, China

**Keywords:** employee stewardship behavior, work-to-family conflict, work–home resources model, work-to-family border permeation, family-supportive supervisor behavior, family support

## Abstract

Drawing on the work–home resources model, our aim in this study was to explore the negative effects of employee stewardship behavior on work–family conflict (WFC) through work-to-family border permeation (WFBP) for employees. A conditional process model linking employee stewardship behavior (ESB), family-supportive supervisor behavior (FBBS), work-to-family border permeation (WFBP), family support, and work–family conflict (WFC) was developed. Longitudinal data collected at two different time points from 323 employees of three internet companies in south China were examined. The results revealed that WFBP mediates the impact of ESB on WFC. Family-supportive supervisor behavior substantially weakens the relationship between ESB and WFBP and the indirect effect of WFBP. Similarly, family support undermines the relationship between WFBP and WFC and the indirect effect of WFBP. Employee-level stewardship and blurred work–family boundaries have been common phenomena in contemporary China, especially during the COVID-19 pandemic. This study is among the first to focus on the negative impacts of employee stewardship behaviors on the employee, especially on their family, from a Chinese context. These findings also increase our understanding of the effects of ESB and provide some new insights into how to mitigate WFC.

## 1. Introduction

Stewardship behavior refers to the attitudes and behaviors of individuals who place the organization’s long-term interests above their own [1]. Stewardship behavior is a pro-organizational behavior motivated by intrinsic motivation [2,3] and is altruistic and voluntary. Previous research on stewardship behavior focused on managers and family firms [4,5]. Employees’ stewardship behavior is prevalent in the Chinese context, where collectivism is emphasized [6]. With increasing organizational competition, stewardship behavior can enhance individual creativity and organizational performance, including creating a competitive advantage [7,8,9]. Moreover, stewardship behavior is beneficial for organizational survival and sustainability [3]. Given the positive impact of stewardship, researchers have studied which factors motivate individual stewardship behavior, including psychological ownership [10] and trust [1] at the individual level; and leadership behaviors [1], collectivism, and low power distance organizational culture [2,11] at the organizational level. However, limited attention has been paid to the possible negative consequences of stewardship behavior on employees and their work–family relationships.

Since 2020, influenced by the ongoing COVID-19 pandemic and closure policies, work from home [12,13] and flexible working arrangements [14,15] have become a passive or an active choice for most companies. Even with the development of communication technologies, working from home has become a regular part of employees’ work lives [15,16]. “If there is hysteresis as people learn new ways to work remotely and businesses reorganize, the pandemic-driven changes may portend more lasting effects on the organization of work” [17]. However, working from home has posed new challenges for companies. Working from home has blurred the work–family boundary and family–work conflicts have become more serious. According to a 2018 NASDAQ: JOBS survey of white-collar workers in China, more than 60% of respondents experienced varying degrees of work–family conflict, which have been exacerbated to some extent by the pandemic since 2020. The recent literature on working from home also reflects the conflicts that have arisen between family and work during the pandemic lockdown, such as increased stress levels [18], household and childcare burdens [19] and work interruption [20], among others. Additionally, due to the employee supervision challenges created by working from home, the initiative and self-motivation of employees had to be enhanced to improve their efficiency and productivity. Employee stewardship, as a voluntary pro-organizational behavior of employees, will play a more important role in work-from-home scenarios. However, the studies in the work–family domain during the pandemic have focused on employee well-being [21], work engagement, and stress [22], but studies of employee stewardship behavior are lacking. In work-from-home scenarios during a pandemic, employees with stewardship behaviors may prioritize the allocation of limited resources to work, reducing family-to-work conflicts and thus safeguarding work efficiency and quality. However, employees’ stewardship behavior toward the organization may exacerbate work–family conflict, having negative consequences for the good stewards as well as their families. Therefore, the impact of stewardship behavior on employee work–family relationships should be explored. Theoretically, the aim of this study was to fill the gap in the research on the family spillover effects of employee stewardship behavior, as well as on stewardship behavior as a predictive factor in the work–family domain during the pandemic. In terms of organizational practice, understanding the potential benefits and costs of employee stewardship behavior on employees can better guide managers in implementing interventions.

The findings of this study contribute to the current literature in three aspects: First, by exploring the negative impact of employee stewardship behavior (ESB) on work–family conflict (WFC), the current understanding of the cost of good stewards in work–family balance situations is understood. Second, given the highly blurred work–family boundary situation during the pandemic, we investigated how good stewards produce WFC via the penetration of work into the family. Third, the findings shed light on the mediation effect of ESB on WFC by exploring the moderating role of two types of support from work and family situations, which extends the boundary conditions of the process of ESB on WFC and advances our understanding of how to help good stewards balance work and family.

## 2. Theoretical Background and Hypotheses

Based on the work–home resources model (W-HR) [23], we argue that employee stewardship has a negative spillover effect on work–home relationships. The W-HR is based on resource conservation theory [24], a theoretical framework that was explicitly proposed to explain positive and negative work–home processes. The W-HR provides an accurate explanatory perspective for research in the work–home domain in terms of resource conservation and depletion [23,24]. The W-HR theory states that WFC occurs when the workplace demands consume personal resources and prevent individuals from contributing to the family domain. As mentioned above, with the increasing virtualization of work and the occurrence of COVID-19, working from home and flexible work arrangements have further contributed to the blurring of work–family boundaries [16,25]. Good stewards are often able to provide valuable personal resources (e.g., time, energy, and other resources) for the collective good of the organization, regardless of personal reward [1]. Good stewards willingly focus more on their work domain, which likely penetrates the family domain, leading to WFC. Based on the current reality and theoretical relevance, in this study, work–family permeation was chosen to represent the resource depletion mechanism.

The W-HR model suggests that the extent to which individual behavior generates or consumes resources varies across situational resource conditions and thus has different effects on work–family relationships [26]. According to the W-HR framework, support from work or family situations can provide additional resources for individuals to better balance and allocate resources in work and family settings, thereby inhibiting the negative effects of employees’ stewardship behaviors on the family domain. Family-friendly organizational policies and informal organizational support are essential resources that enable employees to manage their work and family responsibilities [27,28,29]. Specific support from supervisors in the family domain, such as empathy, resources, and action, can help subordinates achieve a better balance between work and family responsibilities than general organizational support [26,27,30]. In non-work situations, social support, specifically from family members, can help employees avoid work–family conflict [31]. Therefore, we introduced two contextual resources, family-supportive supervisor behavior (FSSB) (support from the organizational situation) and family support (support from the social situation), to examine their moderating effects on the depletion mechanisms of ESB.

In summary, we investigated the mechanism through which ESB influences employees in terms of work–family conflict (WFC). Specifically, we examined the work–family permeation’s mediation effect of ESB on WFC. We explored whether the indirect effect of ESB on WFC through work–family permeation is moderated by two sources of support, namely, the supervisor’s family support and family support. Figure 1 shows the research model.

### 2.1. ESB and WFC

Stewardship behavior refers to managers who are willing to sacrifice their self-interests to maximize the interests of their principals or shareholders and was initially targeted at managers rather than employees [32]. Stewardship behavior has essential value for organizational governance practices and sustainable development [33]. Stewardship behavior was then extended from the subject of managers to employees [1]. This concept refers to the behavior of individuals in an organization (either managers or employees) who prioritize the organization’s long-term interests over their personal short-term interests. ESB comprises two essential characteristics: (1) dedication, where stewards contribute valuable personal resources, such as time, energy, and other resources, to the collective good of the organization, regardless of personal reward [1]; (2) organizational long-term interest priority, where stewards prioritize the collective long-term interest of the organization when they are faced with a conflict between their individual short-term interests and the collective long-term interests of the organization [2]. Particularly in the Chinese context, stewards are more likely to prioritize organizational requirements over their family matters because of individual perceptions of collectivism, which leads to WFC.

Work–family conflict refers to the inter-role conflict that arises when the demands of work and family roles conflict with each other [34]. Work–family conflict can be further divided into two directions: WFC and family–work conflict [35]. We only discussed WFC in this study. Individuals have limited time, attention, and energy (physical and mental) resources [36]. Allocating resources to one role means reducing the resources available for other roles [37]. Good stewards contribute more time, energy, and other resources to the work domain and fewer resources to the family, resulting in a WFC. Based on this, the following hypothesis is proposed:

**Hypothesis** **1.***ESB is positively related to WFC*.

### 2.2. Mediating Role of Work–Family Boundary Permeation

Work–family boundary permeation is generally defined as the extent to which elements of one domain enter another [38]. By contrast, the willingness to cross work–family boundaries is the extent to which an individual’s current role allows for the integration and absorption of the elements of another role [39]. Similarly, work–family penetration can be divided into two aspects: work–family penetration and family–work penetration [38]. In this study, we mainly discussed work–family penetration. Furthermore, permeability is asymmetric [40]; for example, the same individual may have work–family permeability but weak family–work permeability. Permeability depends on the differential energy of the work and family domains [38]: the stronger the domain, the more likely its boundaries are to be firm, so are less likely to be penetrated by other domains but may penetrate other domains.

According to the W-HR model, work roles can prevent individuals from fulfilling their family responsibilities because of limited resources and spillover effects [30]. Stewardship behavior implies that employees voluntarily become stewards and protectors of the organization, pursue their social values, and focus on the long-term and sustainable development of the firm [1]. Thus, individuals with stewardship behavior tend to have a stronger organizational identity [41]. Then, because the work domain of individuals with stewardship behaviors is more important than the family domain, resource allocation skews toward the work domain. This can even occur at the expense of allocating resources on the family side to secure resources on the work side, which more likely produces work–family penetration.

Boundary penetration can be classified into three types: physical, temporal, and psychological penetration [38]. Physical penetration refers to work (or family) matters or people entering the home (or work) space, whereas temporal penetration refers to work (or family) time requiring home (or work) time. Psychological penetration mostly refers to emotional or attitudinal spillover and the migration of ideas and skills. High-permeability boundaries allow the aspects of a person’s characteristic behaviors and expressions to spill over into another life area [38]. In addition, permeability can indicate that an individual’s current role allows them to be in one area when their mind is focused elsewhere (i.e., to be absent of mind, this is thinking about home while at work or thinking about work while at home) [42]. Work penetration into the home (e.g., working from home, taking work home, work-related contacts at home, and work–family multitasking) was associated with work–family conflict and stress [43]. According to the above analysis, individuals with stewardship behaviors are more likely to experience objective versus subjective work–family penetration, and work–family penetration exacerbates work–family conflict [29,44,45]. Therefore, the following hypothesis was constructed:

**Hypothesis** **2.**
*Employees’ WFBP mediates the relationship between ESB and WFC.*


### 2.3. Moderating Role of a Family-Supportive Supervisor and Family Support

According to the W-HR model, we suggest that support from the organization can attenuate the relationship between stewardship and work–family penetration. Support from the family can attenuate the relationship between work–family penetration and WFC. First, workplace social support is a vital work resource that positively impacts individuals and organizations [46,47]. Supervisor-specific support of the family domain is more strongly associated with employees’ work–family conflict than general organizational support [48]. The supervisor behavior that involves the understanding and support of employees in balancing work and family life is defined as family-supportive supervisor behavior (FSSB). FSSB is evidenced by supervisors providing emotional and instrumental support to employees, modeling work–family balance, and implementing innovative work–family management [30]. FSSB supports both work and family and helps employees balance the dual stress of work and family roles, thereby effectively alleviating the work–family conflict faced by employees [49]. When the degree of FSSB is high in an organization, supervisors help employees balance work and family life by providing resources and support and adopting innovative management practices, such as initiating work reorganization and flexible work hours and work locations. Thus, employees gain a degree of job autonomy [30] and self-actualization is promoted [50]. By stimulating positive emotions and encouraging employees to enhance their work skills and improve their productivity, for employees with stewardship behaviors, the possibility of their work penetrating into their families is lower.

Family-supportive supervisors set an example of fulfilling family responsibilities by sharing techniques they use for balancing work and family life with their subordinates [50] and supporting employees in their family responsibilities. As such, stewardship employees can perceive the organization’s support of their own families. They are willing to restrict the penetration of work into their families when responding to their supervisors’ requests. Conversely, when the degree of FSSB in the organization is low, employees with a tendency toward stewardship behaviors are willing to assume more work responsibilities. ESB leads to higher willingness toward and increased work–family penetration. Based on the above analysis, we propose the following hypothesis:

**Hypothesis** **3a.**
*Family-supportive supervisor moderates the positive relationship between ESB and WFBP so that the positive relationship is weaker when the employees perceive that the family-supportive supervising is high.*


In terms of the family situation, family support substantially impacts work–family conflict [31]. The actual work–family conflict occurs between the work and family domain needs [34]. If employees choose to prioritize work needs, it may result in the infiltration of work into the family, which produces WFC. Family support is divided into instrumental and emotional support.

Instrumental support refers to the behaviors and attitudes of family members in managing the family’s daily affairs, which reflects the willingness of family members to share the responsibilities of family affairs. Examples of this support include actively helping those working outside of the home in the family to reduce and eliminate excessive family responsibilities and family affairs and adjusting family life to the work schedule or needs of those who are out working [51]. Strong family instrumental support means that family members proactively share household chores and reduce the objective resource investment required by employees in the family. Thus, the family stress caused by the actual work–family permeation situation decreases (e.g., a short time at home, presence of cared-for members, and separation due to work) [51], thereby promoting higher work–family balance.

Emotional support refers to behaviors and attitudes including encouragement, understanding, concern, greeting, and problem-solving orientation expressed by the family. That is, family members show their willingness to listen to and talk with the employee in the family (outworking) and provide advice to the person outworking about work [51]. For employees with high emotional family support levels, family support allows employees to perceive that their choice to prioritize work is accepted, accommodated, and understood. Particularly in China, a work-first social norm is commonly held, where work is more than just a personal matter and a means to enhance the overall interests and honor of the family [52]. Hard work is also considered a sign of having family responsibilities [52,53]. Family members of Chinese employees tend to express understanding of the infiltration of work matters into the family. This family support behavior may weaken the WFC resulting from work-to-family infiltration. Conversely, when the level of family support in the family is low, the infiltration of work into the employee’s family will lead to subjectively and objectively higher WFC. Based on the above analysis, we propose the following hypothesis:

**Hypothesis** **3b**. *Family support moderates the positive relationship between WFBP and WFC, so that the positive relationship is weaker when the employees perceive that family support is strong.*

The relationship between stewardship behavior and WFC examined in this study reflects the contradiction between the demands of the work and family domains. Given the limitation of personal resources (time and energy), employees with a tendency toward stewardship behavior cannot effectively balance work and family roles. Employees are more willing to prioritize work over family responsibilities, work overtime, or take work home to complete, and their roles penetrate the family domain, causing WFC. According to the W-HR model, FSSB relieves the stress imposed by work demands and helps employees preserve resources in the work and family domains by providing support that balances the demands of both domains. Moreover, FSSB provides employees with a greater sense of control over work and family demands [47] and access to critical work–family conflict resolution resources [54]. The family-supportive leader in the organization emphasizes support of the family aspects of the employee’s life, potentially undermining the WFC caused by ESBs through work-to-family penetration. When the degree of FSSB in the organization is low, employees inclined toward stewardship behavior will experience more pressure from work; sacrifice their personal lives, including family, for the betterment of the organization; and become more unable to balance work and family demands. Therefore, Hypothesis 4a was constructed:

**Hypothesis** **4a.**
*FSSB moderates the indirect effect of ESB on WFC via WFBP, such that the indirect effect is weaker when the employees perceive that the family support of the supervisor is high.*


Employees with a tendency toward stewardship behavior usually experience high-frequency and -intensity work–family permeability. The more permeable the work–family boundaries, the greater the work–family conflicts [29,45]. At strong levels of family support, employees perceive less family stress and they can more efficiently complete work tasks and responsibilities, allowing resources to be effectively preserved while allowing time and energy for family needs, objectively weakening the WFC caused by stewardship behaviors through work–family permeability. When the degree of family support is low, employees with stewardship behavior face pressure from work and family, which more likely causes the depletion of personal resources (e.g., time and energy). Thus, complicated family affairs affect employees’ positive emotions and work efficiency. Employees cannot efficiently complete organizational requirements, so spend more time and energy meeting work demands. For instance, working overtime and working from home after work may cause mutual complaints between employees and families, reinforcing the WFC caused by stewardship behavior through work-to-family penetration. Based on the above analysis, we propose the following:

**Hypothesis** **4b.***Family support moderates the indirect effect of ESB on WFC via work–family border permeation, such that the indirect effect is weaker when the employees perceive that family support is high*.

## 3. Methods

### 3.1. Procedure and Participants

In September 2020, we collected survey data from 323 employees at two stages from 3 internet enterprises in southern China. Internet enterprises provided an ideal sample for the study because the competition in the industry was more intense, each company had a higher-intensity workload, and work-to-family conflict was more common. In addition, all three internet companies, with approximately 200 employees each, had in-house mobile office systems. All three companies have telecommuting systems and in the second half of 2020, employees were working from home due to the outbreak lockdown. So employees faced a scenario where work–home boundaries were blurred. We used a randomized cluster sample to select qualified departments from the companies, and distributed an anonymous self-assessment questionnaire to all selected respondents (employees). We contacted the HR executives of each of the three companies in advance and created electronic questionnaires through Questionnaire Star, which were sent by the HR executives to the companies’ WeChat groups for data collection. We performed attention checks in both rounds of the survey to ensure that respondents paid attention to all items of the questionnaire.

One set of questionnaires was distributed in our two-phase survey to minimize common method bias. At time 1, we invited 387 employees to evaluate their stewardship behaviors and the level of their supervisors’ family-supportive behavior. We received 365 responses from employees. Two weeks later, at time 2, we invited the 365 participants who had participated in the time-1 survey to report their perceptions of work-to-family border permeation (WFBP), family support, and WFC. Of the 365 participants, 341 responded. After the data collection, we filtered and deleted the problematic ones (e.g., incompletely answered questionnaires). The valid sample consisted of 323 employees from three organizations, resulting in a final response rate of 83.5%. Among the 323 employees, 56.97% were female, 80.19% had a bachelor’s degree and above, 75.23% were aged under 35 years, the average age was 31.460 years (SD = 9.581), and the length of working years was mainly distributed within 1–10 years (75.23% of the total). The ratio of employees without children to those with children was 6:4.

### 3.2. Ethical Considerations

The study complied with the Declaration of Helsinki and followed its ethical codes for individuals, samples, and data collection involved in each research procedure. Before the initiation of this study, we presented the study topic to the Ethics Committee of the School of Business Administration of the South China University of Technology and submitted a proposal stating the purpose of the study, sample, data sources, and details of the written informed consent for participants. All of the above documents were approved by this committee. Prior to the questionnaire, the researchers asked the participants to read the written informed consent carefully, introduced the purpose of the study to the participants, and explained that the data would be used for research only and that all information about the participants would be kept confidential. All participants were informed and volunteered to complete the questionnaire.

### 3.3. Measures

To ensure the scale equivalence in Chinese, we followed the translation and back-translation procedure to translate the original scales. All items were measured on five-point Likert scales (1 = strongly disagree, 5 = strongly agree).

ESB: ESB was measured with the three-item scale adapted by Davis, Allen, and Hayes [32]. An example item is “I have initiatives that serve the company’s interests more than my own.” In this research, Cronbach’s alpha reliability coefficient was 0.823.

WFBP: We used the six-item scale developed by Clark [38] to measure employees’ willingness to work to penetrate family. An example item is “I will think about work-related concerns while I am at home.” In this research, Cronbach’s alpha reliability coefficient was 0.872.

WFC: The WFC was assessed using the five-item scale of Netemeyer [55]. An example item is “Things I want to do at home do not get done because of the demands my job puts on me.” In this research, Cronbach’s alpha reliability coefficient was 0.904.

FSSB: We used Hammer’s four-item family-supportive supervisor scale to measure employees perceived family support from their supervisor [30]. The scale contains four dimensions: emotional support, instrumental support, role modeling behavioral, and creative work–family management. The sample statements include “Your supervisor works effectively with employees to creatively solve conflicts between work and nonwork.” The reliability for the overall scale was 0.893.

Family support: Employee-perceived family support was measured with a 10-item scale [51]. A sample item is “When I am busy at work during a certain period, my family always does more housework.” The reliability for this scale was 0.903.

Control variables: Following prior studies [40], we controlled several variables at the individual level. Specifically, we controlled for some demographic variables (i.e., gender, age, education, employee working length). In addition, we controlled whether employees have children and the number of children they have because this will impact the resources they need to invest in the family.

## 4. Results

### 4.1. Preliminary Analyses

Common method bias test. In order to reduce the effect of common method bias, this study specifically controlled for both measurement procedures and statistical methods. For the measurement procedure, the researcher used a multi-temporal method in which questionnaires were administered to the same employees at two points in time (2 weeks apart) and the measurement items in the questionnaires were randomly arranged; in order to reduce employees’ guesses about the measurement items, the researcher used anonymity for all participating employees. For statistical methods, this study tested for common method bias by performing Harman’s one-way test with IBM SPSS Statistics 22.0 software (Shanghai, China). Five factors with eigenvalues greater than one were analyzed, and none of them explained most of the variance. The cumulative explained variance of these factors was 64.865%, and the maximum variance contribution of the common factor was 24.019%. Therefore, there was no significant common method bias in the study data.

Descriptive statistics and correlation analysis. Table 1 reports the means, standard deviation, Cronbach’s alpha coefficients of the variables, and their correlation coefficients. In line with our expectations, ESB was positively associated with WFC (r = 0.173, *p* < 0.01). The reliability coefficients of all measures were above 0.80, thereby passing the 0.70 threshold that is considered acceptable for research use.

Confirmatory factor analysis. We conducted confirmatory factor analysis (CFA) to examine whether the self-report measures captured specific constructs and standard method bias existed. Table 2. shows the CFA results. The results showed that the five-factor model provided a better fit to the data (χ^2^ (340) = 565.447, TLI = 0.948, CFI = 0.954, RMSEA = 0.045, SRMR = 0.044) than alternative models and passed the model fit indices thresholds suggested by Hu and Bentler [56], verifying the distinctiveness of our measures.

### 4.2. Hypotheses Testing

We used SPSS22.0 to test the theoretical hypothesis. Hypothesis 1 predicted a positive relationship between ESB and WFC. As shown in Table 3 Model 5, ESB had a significant positive effect on WFC (β = 0.238, *p* < 0.01). Thus, Hypothesis 1 was supported. Further, in Model 6, the effect of ESB on WFC was reduced and not significant. Meanwhile, WFBP was significantly related to WFC (β = 0.416, *p* < 0.001), indicating that WFBP mediated the influence of ESB on WFC. In addition, we used the Monte Carlo simulation approach [57] to assess the indirect effect. The results showed that the indirect effect of ESB on employees’ WFC was 0.238, 95% CI = [0.095, 0.380], reaching a significant level. Therefore, Hypothesis 2 is supported.

The data analysis results of Model 4 in Table 3 show that the interaction term between ESB and FSSB has a significant negative effect on WFBP (β = −0.149, *p* < 0.01), which is inconsistent with the direction of ESB’s effect on WFBP (β = 0.367, *p* < 0.001). Therefore, under the moderation of FSSB, the effect of ESB on EFBP was reduced (β = 0.321, *p* < 0.001). Hypothesis 3a is supported.

According to the data analysis results of Model 7 in Table 3, the interaction term between WFBP and family support can significantly negatively affect employees’ WFC (β = −0.241, *p* < 0.001), which is inconsistent with the direction in which WFBP affects employees’ WFC (β = 0.416, *p* < 0.001). Therefore, under the moderating effect of family support, the effect of WFBP on employee WFC was weakened (β = 0.37, *p* < 0.001). Hypothesis 3b is supported.

Figure 2 further illustrates that FSSB plays an interfering moderating effect in the process of ESB, positively affecting WFBP. Under low FSSB, the regression slope of ESB on WFBP is relatively inclined; under the condition of high FSSB, the regression slope is relatively flat and low.

The moderated path analysis approach [58] was applied to test the moderated mediation relation that FSSB would moderate the indirect effect of ESB on WFC through WFBP. The results revealed that FSSB significantly moderated this indirect effect (difference = −0.125, *p* < 0.05, 95% CI = [−0.238,−0.031], excluding 0). Specifically, when FSSB was low (one SD below the mean), the moderated indirect effect was 0.1337 (*p* < 0.001, 95% CI = [0.0592, 0.2205], excluding 0); when FSSB was high (one SD above the mean), the moderated indirect effect was 0.0743 (n.s., 95% CI = [−0.025, 0.1767], including 0). Thus, Hypothesis 4a was supported.

Hypothesis 4b predicted that the indirect effect of ESB on WFC via WFBP would be moderated by family support. The results showed that this effect was significantly moderated (difference = −0.123, *p* < 0.01, 95% CI = [−0.213, −0.054], excluding 0). Specifically, when family support was high (one SD above the mean), the moderated indirect effect was 0.075 (*p* < 0.1, 95% CI = [0.007, 0.171], excluding 0); when family support was low (one SD below the mean), the moderated indirect effect was 0.197 (*p* < 0.001, 95% CI = [0.112, 0.298], excluding 0). Thus, Hypothesis 4b was supported.

As shown in Figure 3, the diagram of the interactive effect of family support is consistent with Hypothesis 3b. When the degree of family support is low, the WFC appears to be higher; when the degree of family support is high, the WFC appears to be lower. Given that the absence of resources is more efficient than resource acquisition, the absence cycle has a more significant influence and faster rate of change on the acquisition cycle [24].

The path coefficient diagram obtained by Mplus8.3 similarly verifies these hypotheses (see Figure 4).

## 5. Discussion

As previous researchers focused on the positive outcomes of stewardship behavior and its influencing factors, the impact of stewardship behavior on employees themselves was previously unclear. In this study, we investigated how employee stewardship affects employees’ work–family relationships through the resource consumption mechanism (work–family permeability) based on the W-HR model. Moreover, we examined how the contextual characteristics (support from organizational and family situations) moderate the different mediating mechanisms of various factors such as ESB and blurred work–family boundaries are prevalent in the Chinese context. The results showed that ESB negatively affected work–family relationships through work-to-family permeation. In addition, the relationship between employee stewardship and work-to-family penetration was moderated by family-supportive leadership behaviors; the relationship between work-to-family penetration and WFC was moderated by family support.

### 5.1. Theoretical Implications

Our findings provide a fourfold contribution to the literature. First, the findings advance the understanding of the effects of employee stewardship behaviors. Previously, researchers typically focused on how stewardship behavior benefits employees, organizations, and stakeholders [59]. Thus, the influence of stewardship behavior has been studied more in terms of individual- and organizational-level factors, such as psychological, structural, supervisory, situational, and cultural factors [10], with a lack of focus on the possible negative consequences of stewardship behavior. As a voluntary pro-organizational behavior, employees who exceed work expectations are prone to fatigue, which has conceptualized the negative fatigue-related consequences of employee involvement in organizational citizenship behavior as organizational citizenship fatigue [60]. These findings help us understand that good stewardship comes at a cost as individuals have limited time, energy, and resources. We explored the negative effects of employee stewardship behavior on employees based on an actor-centered perspective. Based on the W-HR model, we explored the positive and negative effects of employee stewardship behavior on employees in terms of the mechanism of resource consumption (the penetration of work into the family). We found that good stewards allocate more of their limited resources to work and less to family, which helps to more comprehensively explain the positive effects of stewardship behavior on work and the negative effects on employees, and enriches the current literature in the fields of stewardship behavior and work–family boundaries.

Second, our findings enrich the employee stewardship behavior and work–family relationship literature. Most previous studies on stewardship behavior have been limited to the work domain [61], and the spillover effects of stewardship behavior on the family have been ignored. However, with the development of communication technology and the arrival of the post-pandemic era, employees are increasingly working from home, and the boundary between employees’ work and family is being increasingly blurred [25]. Based on this, we focused on the effect of stewardship behavior on employee work–family relationships. The study reveals the mechanism of the effect of employee stewardship behavior on work–family conflict through work-to-family penetration. Based on the W-HR model, this study reveals the “black box” of family spillover effects of employee stewardship from a resource perspective, and organically integrates the study of employee stewardship with work–family relationships.

Third, our study extends the literature on the family cost of good stewardship by uncovering the boundary conditions of two sources of support: work and family. Researchers have mainly explored the direct impact of organizational support versus family support on work–family conflict. This is the first study related ESB to the concept of the family domain; however, according to the W-HR model, the external resource permeation between work and family situations may impact ESB through the process of work-to-family permeation, thereby promoting WFC. We chose FBBS from the work situation and FS from the family situation as the factors affecting the work–family balance relationship. We also investigated its moderating effect on the relationship between ESB and WFC. The results revealed that emotional and instrumental support from FBBS considerably inhibited the willingness to accept and the possibility of work-to-family penetration because of ESB. Emotional and instrumental support from the family strongly inhibited WFC because of employee work-to-family penetration, extending the boundary conditions of the negative effect of ESB. Moreover, the findings advance the current understanding of how good stewards can achieve work–family balance.

Finally, this study expands the research objects and contexts of stewardship behavior. Previously, researchers of stewardship behavior focused on managers and family firms [4], whereas we followed Davis [32] and Hernandez [1] in extending the consideration of stewardship behavior to the employee level. Moreover, researchers on stewardship behavior have focused on the factors and results of stewardship behavior in the workplace. In this study, we considered the possible adverse effects of good stewards’ dedication to the organization on families. During the COVID-19 pandemic, most companies had to adopt more flexible work arrangements and work-from-home patterns, increasing the blurring of work–family boundaries. In a general context, we placed ESB in a dual work–family context and examined the possible adverse effects of ESB in the workplace context on the family. We also considered the moderating effect of support from the work and family domains on this impact path.

### 5.2. Practical Implications

Stewardship is a pro-organizational behavior that contributes to the long-term well-being of the organization’s stakeholders and considerably benefits organizational development and innovation. Especially in a work-from-home scenario during a pandemic, a high degree of conscious pro-organizational behavior (ESB) can effectively disrupt the family. Good stewards have a positive effect on work, but have a negative effect on themselves, and at a cost to the family. Therefore, in practice, managers need to help employees balance work and family by designing and implementing policies and measures to motivate employees to consistently demonstrate balanced stewardship behavior. The results of this study have several implications for managers:

First, managers can provide employees who engage in stewardship behaviors with material and psychological resources. For example, organizations can help employees replenish resources used for practicing stewardship by including ESBs in the scope of performance evaluations and offering financial rewards. Managers can also use various opportunities during the workday to provide employees moral rewards (e.g., recognition and praise) for engaging in stewardship behaviors. Thus, employees may feel a sense of value and meaning by engaging in stewardship behaviors, which can be supported by the organization [62] and can quickly replenish employees’ psychological resources, alleviating the disadvantages of ESBs for employees.

Second, managers can develop support activities that directly or indirectly target employee families. Examples include additional family-related benefits created for employees that allow employees to fully assume their family responsibilities. Moreover, paid family leave, maternity leave, travel allowances, flexible benefits, childcare, early childhood-related allowances, and family days [63] are also helpful. In China, the more common benefits are maternity leave, flexible benefits, family days, and measures to help employees with stewardship behaviors to solve family problems, such as breastfeeding rooms, babysitting rooms, psychological counseling, childcare centers in the workplace, and solutions to children’s schooling problems. More family support is required for employees with stewardship behaviors to mitigate the negative effects on families.

Third, in the recruitment and selection process, a survey can be conducted on employees’ family status to select employees with strong family support. After employees join the company, through training and family concept promotion, they can be supported in establishing harmonious family relationships to relieve their worries about engaging in family-related behaviors, which can more strongly promote work–family balance and ultimately achieve a win-win situation for both the organization and the employees.

### 5.3. Limitations and Future Directions

With this study, we explained the negative effects of stewardship behavior as a whole, but some limitations need to be explored further in future studies:

First, we measured the variables used in this study through employees’ subjective evaluations. Bias in the testing effect was introduced by the possible tendency of employees to protect their privacy and manage their reputation. In the future, researchers can further validate the robustness of the findings through experimental methods or by collecting paired data. In addition, we used a time-lagged approach to obtain data, which reduced the bias of the common approach to some extent but, causal relationships cannot be inferred between variables. Therefore, longitudinal measures should be used in the future to effectively verify causal relationships.

Second, we mainly used a sample of small and medium enterprises (SMEs) from south China. Using this sample effectively controlled the effects of firm type and regional factors and improved the study’s internal validity. However, it also limited the possibility of the findings being extrapolated to other regions of the country or abroad. As the Chinese context is characterized by a high collectivist orientation [6], employee-level stewardship is widespread. In the future, researchers can extend our method study to other regions or empirically examine the effects of employee-level stewardship in other countries. In addition, our study sample was mainly obtained from Internet companies. We did not consider other industries (e.g., employees of state-owned enterprises or nonprofit organizations who may have higher stewardship behaviors) and different types of employees (e.g., managerial versus non-managerial employees), which may have impacted our findings. However, given the lack of consideration of these issues at the beginning of the study design, controlling these control variables during the data analysis was impossible. In future studies, researchers can use data from different industries and other types of employees to reduce the impact of these additional variables and further validate our results.

Third, in exploring the outcome variables of stewardship behaviors, we focused on the effects of ESBs on the family domain and other paths of stewardship behaviors on the family domain, which deserve further exploration. Servant leadership behavior comprises a servant, humble attitude that creates a safe and friendly work atmosphere, allowing leaders to experience positive emotions and therefore smoother and lower-friction daily interactions with subordinates. The positive emotions experienced by leaders after serving their subordinates are a vital personal resource that can have positive spillover effects and moderate work–family conflict [34]. ESBs and servant leadership emphasize selflessness in the organization. Employees who exhibit stewardship behaviors may also experience positive emotions that positively impact WFC. In the future, researchers can examine the positive effects of stewardship behaviors on the family domain.

Fourth, in this study, we mainly examined the impact of ESBs in the work domain on the family domain to explore how good corporate stewardship enables work-to-family balance. However, work-to-family penetration and WFC involve two-way relationships, that is, work-to-family penetration (conflict) and family-to-work penetration (conflict). Additionally, we only considered the mechanism of action in a single direction. In the Chinese scenario, hard work and dedication to the organization are seen as signs of responsibility to the family [52], and stewardship behaviors may also influence the family’s evaluation of the employee and thus family-to-work conflict. The mechanisms of the role of stewardship behavior on the outcomes of both directions of work–family balance (family-to-work vs. work-to-family) can be considered.

## 6. Conclusions

With the ongoing COVID-19 Pandemic, firms to introduce working from home on a large scale. As a voluntary pro-organizational behavior of employees, employee stewardship behavior (ESB) will play a more critical role in work-from-home scenarios. Although good stewards will effectively reduce family interference with work because they will prioritize their limited resources to work rather than family matters, they will create work-family conflicts (WFC) meanwhile. Drawing on the work-home resources model, we proposed a model clarifying how ESB affects WFC. The results revealed that work-to-family border permeation (WFBP) mediates the impact of ESB on WFC. Two sources of support, namely, the supervisor’s family support and family support, weaken the indirect effect of ESB on WFC through WFBP. This study is among the first to focus on the costs of employee stewardship behavior on the employee themselves in the work-to-family penetration context during the COVID-19 Pandemic.

## Figures and Tables

**Figure 1 ijerph-19-06117-f001:**
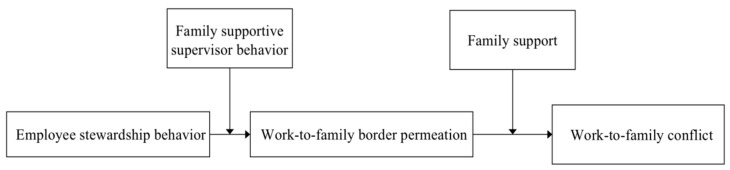
Conceptual research model.

**Figure 2 ijerph-19-06117-f002:**
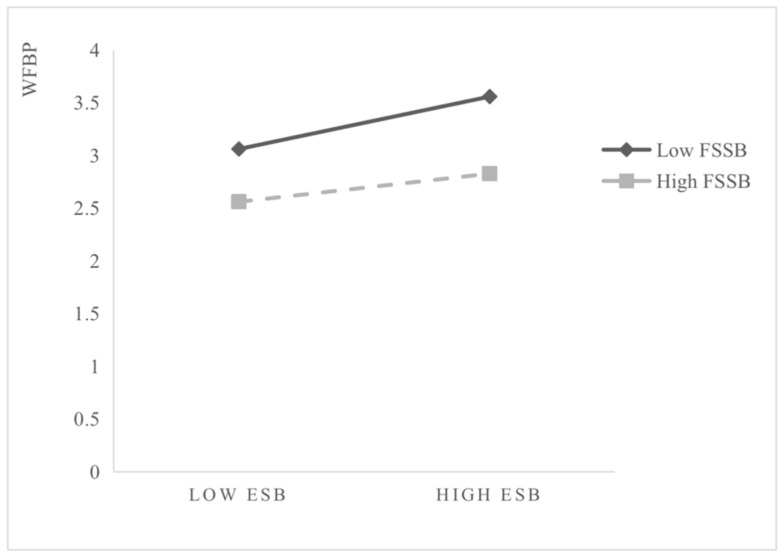
Interaction effect of FSSB.

**Figure 3 ijerph-19-06117-f003:**
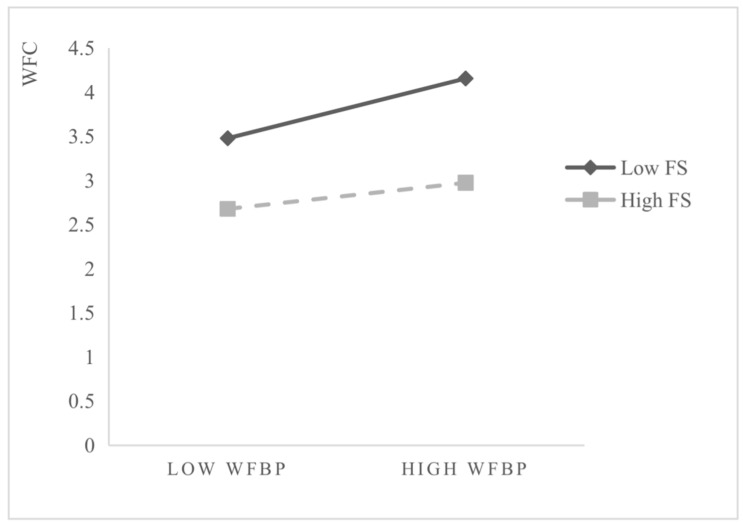
Interaction effect of family support.

**Figure 4 ijerph-19-06117-f004:**
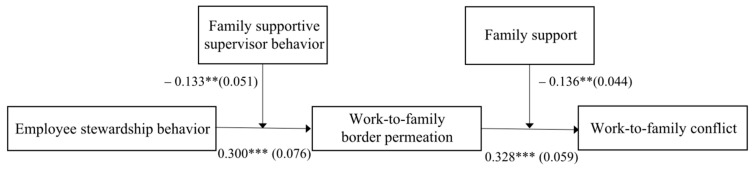
Path coefficient analysis model. Note(s): For brevity and clarity, only the main path coefficients of the full model (unstandardized) are presented in this figure. ** *p* < 0.01, *** *p* < 0.001.

**Table 1 ijerph-19-06117-t001:** Means, standard deviations, and correlations.

	M	SD	1	2	3	4	5	6	7	8	9	10
1. Gender	1.570	0.496										
2. Age	31.460	9.581	0.023									
3.Education	3.110	0.953	0.044	−0.263 **								
4.Children	1.520	0.736	0.049	0.678 **	−0.349 **							
5.Tenure	2.570	1.287	0.028	0.809 **	−0.241 **	0.656 **						
6.ESB	4.041	0.776	−0.021	0.060	−0.125 *	0.116 *	0.110 *	(0.823)				
7.WFBP	3.829	0.830	0.018	0.104	0.088	0.026	0.147 **	0.333 **	(0.872)			
8.WFC	2.971	1.002	−0.074	−0.012	0.033	−0.047	−0.014	0.173 **	0.356 **	(0.904)		
9.FSSB	3.454	0.960	0.030	−0.007	−0.013	−0.006	−0.026	0.154 **	0.042	−0.280 **	(0.893)	
10.FS	3.642	0.759	0.007	0.050	−0.051	0.040	0.038	0.383 **	0.208 **	0.022	0.104	(0.903)

Note(s): Sample size = 323. ESB = Employee stewardship behavior; WFBP = Work-to-family border permeation; WFC = Work-to-family conflict; FSSB = Family-supportive supervisor behavior; FS = Family support. The Cronbach’s alpha coefficients are reported in diagonal. * *p* < 0.05. ** *p* < 0.01 (two-tailed).

**Table 2 ijerph-19-06117-t002:** Confirmatory factor analysis results.

Model	χ^2^	df	χ^2^/df	CFI	TLI	RMSEA	SRMR
Five-factor model	565.447	340	1.633	0.954	0.948	0.045	0.044
Four-factor model ^a^	887.155	344	2.579	0.888	0.877	0.070	0.072
Three-factor model ^b^	1676.689	347	4.832	0.727	0.702	0.109	0.128
Two-factor model ^c^	2400.419	349	6.878	0.578	0.543	0.135	0.253
One-factor model	3481.691	350	9.948	0.356	0.305	0.166	0.184

Note(s): ^a^ Employee stewardship behavior and family support were combined into one factor; ^b^ Employee stewardship behavior, work-to-family border permeation and work-to-family conflict were combined into one factor; ^c^ Employee stewardship behavior, work-to-family border permeation, family-supportive supervisor behavior and work-to-family conflict and were combined into one factor.

**Table 3 ijerph-19-06117-t003:** Moderating effect of FSSB and family support.

Variables	Work-to-Family Border Permeation (WFBP)			Work-to-Family Conflict (WFC)			
	Model 1	Model 2	Model 3	Model 4	Model 5	Model 6	Model 7
b	3.255 ***	1.707 ***	2.013 ***	3.164 ***	2.162 ***	1.452 **	3.222 ***
Gender	0.019	0.031	0.027	−0.148	−0.14	−0.153	−0.122
Age	0.015	0.055	0.052	0.031	0.057	0.034	0.029
Education	0.098	0.127 **	0.117 *	0.026	0.045	−0.008	−0.021
Children	−0.102	−0.133	−0.137	−0.085	−0.105	−0.05	−0.025
Tenure	0.142 *	0.109	0.108	0.008	−0.012	−0.058	−0.067
ESB		0.367 ***	0.321 ***		0.238 **	0.085	0.093
FSSB			−0.014				−0.351 ***
ESB × FSSB			−0.149 **				−0.088
WFBP						0.416 ***	0.37 ***
FS							−0.106
WFBP × FS							−0.241 ***
F	2.764 *	9.728 ***	8.678 ***	0.556	2.281 *	7.43 ***	11.417 ***
R^2^	0.042	0.156	0.181	0.009	0.042	0.142	0.288
Adjust R^2^	0.027	0.14	0.16	−0.007	0.023	0.123	0.262
R^2^ Change	0.042	0.114	0.025	0.009	0.033	0.1	0.146

Note(s): Sample size = 323. ESB = Employee stewardship behavior; WFBP = Work-to-family border permeation; WFC = Work-to-family conflict; FSSB = Family- supportive supervisor behavior; FS = Family support. * *p* < 0.05, ** *p* < 0.01, *** *p* < 0.001.

## Data Availability

The datasets used and analyzed in the current study are available from the corresponding author upon reasonable request.

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
