# Peer review of "Costs of Employee Stewardship Behaviors for Employees in the Work-to-Family Penetration Context during the COVID-19 Pandemic"

_ijerph, 2022, doi:10.3390/ijerph19106117_

Round 1

Reviewer 1 Report

Overall, I think the paper provides a convincing argument and explanatory model based on empirical survey, with a few suggestions for revision.

  1. Since the outbreak of the Covid-19 pandemic, many papers on home-office and work-to-family conflict have been published in the international academia. There are also several publications on China in this arena. It is recommended that authors cite more about WFC and WFBP since 2020/21 and then specifically explain the differences and main innovations between this paper and other published papers.
  2. I would like to know why the title is "the dark side...", instead of directly using the "negative effects..."? The author can explain it to me directly.
  3. Some expressions in this paper are still very similar to Chinglish, and an English native speaker is required to do both semantic and stylistic proofreading.

Author Response

亲爱的审稿人,

感谢您为改进手稿提供建设性建议。我们已根据您的建议修改了论文。所有更改都使用修订中的“跟踪更改”进行标记。

请在附件部分找到我们对您的评论的回复。

此致,

作者

Reviewer 2 Report

Interesting paper - potentially useful contribution given the case of China.  A few comments/points of inquiry:

Section 3 - as there is likely not great understanding of these organizations - what else can you tell us about them?  This is a vague description - how many such organizations are there?  How many people work for them?  What is the work context like?  And so on.  You mention that the work-family boundary is more blurred...more blurred than what?

>>were coded in such a way to guarantee anonymity

Meaning what?

Was the survey protocol subject to any oversight by your university?  If so, what was the protocol to protect research subjects?  Barring this, what specific protections did you afford research subjects?

Your points about stewardship coming at a cost within families are well-taken, especially in the difficult context of a global pandemic. 

With respect to the implications of your findings, though, you mention that according flexibility to employees is appropriate, as well as allowing for different forms of paid leave and rewards.  This seems fairly obvious though.  Are there reasons why employers don't do this?  Do they already do this and it simply is not effective?  Do they try these approaches and simply do not sustain the effort?  This section is not fully realized; a consideration of employee-support efforts these companies have already engaged is not included here, and it seems like an assumption that companies have not done enough.  They may not have done enough, but I would rather not assume that.

Author Response

亲爱的审稿人,

感谢您为改进手稿提供建设性建议。我们已根据您的建议修改了论文。所有更改都使用修改中的“跟踪更改”标记。

请在附件部分找到我们对您评论的回复。

此致,

作者

Reviewer 3 Report

Dear Author,

Thank you foe effort in this work.

Kindly see the following notes:

Emphasis should be placed on a theory that explains the supposed relationships between variables.

Justify the reason for choosing 3 companies for the Internet and in south China.

What type of sample was used?

Add the resulting shape from the structural equation

What is the justification for the development of five models for analysis?

Show the real contribution of this research in the discussion section.

Best Regards,

Author Response

(The authors gave the same response as above.)

Round 2

Reviewer 2 Report

The authors have addressed my concerns acceptably.